# T Regulatory Cell-Associated Tolerance Induction by High-Dose Immunoglobulins in an HLA-Transgenic Mouse Model of Pemphigus

**DOI:** 10.3390/cells12091340

**Published:** 2023-05-08

**Authors:** Christoph Hudemann, Jochen Hoffmann, Enno Schmidt, Michael Hertl, Rüdiger Eming

**Affiliations:** 1Department of Dermatology and Allergology, Philipps-University Marburg, 35037 Marburg, Germany; 2Department of Dermatology, University of Heidelberg, 69117 Heidelberg, Germany; 3Department of Dermatology, University of Lübeck, 23562 Lübeck, Germany; 4Lübeck Institute of Experimental Dermatology (LIED), University of Lübeck, 23562 Lübeck, Germany; 5Department of Dermatology, Venerology and Allergology, German Armed Forces Central Hospital Koblenz, 56072 Koblenz, Germany

**Keywords:** pemphigus vulgaris, desmoglein, IVIg, antibodies, autoimmunity

## Abstract

Pemphigus vulgaris (PV) is a potentially lethal autoimmune bullous skin disorder caused by IgG autoantibodies against desmoglein 3 (Dsg3) and Dsg1. During the last three decades, high-dose intravenous immunoglobulins (IVIgs) have been applied as an effective and relatively safe treatment regime in severe, therapy-refractory PV. This prompted us to study T- and B- cell polarization by IVIg in a human-Dsg3-dependent mouse model for PV. Using humanized mice transgenic for HLA-DRB1*04:02, which is a highly prevalent haplotype in PV, we employed IVIg in two different experimental approaches: in prevention and quasi-therapeutic settings. Our data show that intraperitoneally applied IVIg was systemically distributed for up to 42 days or longer. IVIg-treated Dsg3-immunized mice exhibited, in contrast to Dsg3-immunized mice without IVIg, significantly less Dsg3-specific IgG, and showed induction of T regulatory cells in lymphatic tissue. Ex vivo splenocyte analysis upon Dsg3-specific stimulation revealed an initial, temporarily reduced antigen-induced cell proliferation, as well as IFN-γ secretion that became less apparent over the course of time. Marginal-zone B cells were initially reduced in the preventive approach but re-expanded over time. In contrast, in the quasi-therapeutic approach, a robust down-regulation in both spleen and lymph nodes was observed. We found a significant down-regulation of the immature transitional 1 (T1) B cells in IVIg-treated mice in the quasi-therapeutic approach, while T2 and T3, representing a healthy stage of B-cell development, appeared to be up-regulated by IVIg. In summary, in two experimental settings employing an active PV mouse model, we demonstrate distinct alterations of T- and B-cell populations upon IVIg treatment, compatible with a tolerance-associated polarization in lymphatic tissue. Our data suggest that the clinical efficacy of IVIg is at least modulated by distinct alterations of T- and B-cell populations compatible with a tolerance-associated polarization in lymphatic tissue.

## 1. Introduction

Pemphigus is a potentially lethal autoantibody (auto-ab)-driven autoimmune disorder involving mucous membranes and the skin [1,2]. In general, immunoglobulin 4 (IgG4) Abs are predominantly found in pemphigus sera of patients with active disease, followed by IgG1 and occasionally IgG2 and IgG3. PV can be classified into two major subtypes, depending on the observed auto-ab profile. Desmoglein 1 (Dsg1)-specific auto-abs induce subcorneal blister formation characteristic of pemphigus foliaceus [3]. In pemphigus vulgaris (PV), Dsg3-specific antibodies induce acantholysis in the basal and suprabasal layers of the mucous membranes, resulting in painful and slow-healing sores [4,5]. The current Dsg3/Dsg1 compensation theory states that Dsg3 compensates for the loss of Dsg1, leading to clinically active anti-Dsg1 in the skin, whereas anti-Dsg3 IgG leads to impairment of mucosal epidermal adhesion due to the low expression of Dsg1 that cannot adequately compensate the loss of Dsg3 adhesion [4]. The annual incidence rate of PV was found to be between 0.8 and 16.1 per million population, depending on ethnicity and geographical area [6]. In addition to the main clinical symptoms, compelling comorbidities were found between pemphigus and other autoimmune disorders, ranging from rheumatoid arthritis, psoriasis and different malignancies to neurologic diseases [7]. Additionally, IgG against several non-Dsg3 proteins, such as desmocollin 3, has been detected in pemphigus patients, raising speculation of potential synergic effects eventually triggering acantholysis [8,9,10].

Treatment of pemphigus largely relies on systemic corticosteroids (SCs) and non-specific immune suppressants [10,11]. The current guidelines recommend systemic high-dose corticosteroids combined with immunosuppressive agents, such as mycophenolate mofetil, azathioprine or the anti-CD20 monoclonal antibody, rituximab [12].

Intravenous immunoglobulins (IVIgs) refers to the intravenous application of highly concentrated human immunoglobulin G [13]. Standard IVIg products are antibody concentrates derived from thousands of plasma donations from healthy volunteers, thus representing the broad spectrum of antibodies in human blood. They are an important treatment option in antibody replacement therapy. Because of their immunomodulatory properties, they are also used in various autoimmune and inflammatory disorders [14,15]. Antibodies against self and foreign antigens are present in IVIg and may lead to a rapid and selective decline in serum levels of PV-associated auto-ab [16] and a subsequent improvement in the clinical picture (Appendix A). While the most likely main mechanism of action of IVIg is blockage of the neonatal Fc receptor, a plethora of additional effects has been described, including both Fc- and F(ab)_2_-mediated effects [17]. Its various anti-inflammatory effects, however, can be associated in part with an inhibition of cytotoxic CD8+ T cells [18], while inducing CD4+ T regulatory cells [19]. Additionally, the induction of B-lymphocyte apoptosis, inhibition of phagocytosis and increment of response to corticosteroids contribute to the broad function of IVIg with respect to auto-inflammatory diseases, such as PV [20].

Previous studies have shown the protective effects of IVIg in experimental mouse models of PV [21] and further autoimmune bullous diseases, such as experimental epidermolysis bullosa acquisita [22,23]. Several murine pemphigus models are available (active, passive), each facilitating the analysis of a characteristic feature, such as pathogenic IgG or Dsg3-specific T or B cells [24]. We and others have shown that PV-associated HLA class II alleles, such as HLA-DRB1*04:02, are involved in the activation of Dsg3-specific auto-aggressive CD4+ T cells via the presentation of auto-antigenic peptides, leading to the induction and maintenance of autoreactive memory B cells [25,26].

In the present project, we aim to analyze the Dsg3-dependent T- and B-cell polarization and the modulating effects of intraperitoneally applied IVIg in an HLA-DRB1*04:02-transgenic mouse model of PV.

## 2. Materials and Methods

*Protein production and purification*—Extracellular domain of human Dsg3 and the unrelated control protein, type XVII collagen (non-collagenous 1, NC1), were produced in baculovirus-infected insect cells (High Five; Invitrogen, Carlsbad, CA, USA) as described previously [27,28]. Proteins were purified with affinity chromatography using nickel–nitrilotriacetic agarose beads (Qiagen, Hilden, Germany) according to the manufacturer’s instructions. Purified protein was gradually dialyzed against PBS supplemented with 0.5 mmol/L CaCl_2_. Quality control was performed with Coomassie staining and ELISA using patient sera.

*Immunization of HLA-DR4-transgenic mice and IVIg treatment*—8- to 12-week-old HLA-DRA1*01:01-DRB1*04:02/-DQA1*03:01, -DQB1*03:02 (DQ8)-transgenic C57Bl/6J male and female mice expressing the human CD4 co-receptor and deficient in I-Aβ (I-Aβ−/−) were immunized using intraperitoneal (i.p) injection with recombinant human Dsg3(1-566) extracellular domain (40 μg) in alum on day 0, followed by immunization with 40 μg Dsg3 on days 14 and 28 (group A2, A3) [25]. Human high-dose immunoglobulin infusion solution (Intratect^®^, Biotest Pharma GmbH, Dreieich, Germany) was administered weekly intraperitoneally at a dose of 2 g/kg at indicated time points (Figure 1). Analysis was performed focusing on blood and lymphatic tissue (i.e., spleen, inguinal and lumbar lymph nodes).

*ELISA*—For serum antibody quantification of the immunized mice, a similar recombinant human Dsg3(1-566) protein was coated onto immune-microtiter plates overnight at 4 °C (96-Well; Greiner Bio-One, Frickenhausen, Germany). After blocking with a blocking buffer (PBS + 5% milk powder) for 2 h, sera from Dsg3-immunized mice served as a source of primary antibodies at a dilution of 1:400 for 2 h at room temperature. For visualization, HRP-conjugated anti-murine antibodies (1:2000, Dako, Glostrup, Denmark), followed by development using ABTS (Sigma, Saint Louis, MO, USA), were used. Absorbance level was measured at 405 nm (Tecan plate reader Sunrise + Magellan software (V7.2); Tecan Group Ltd., Männedorf, Switzerland).

*Functional ex vivo analysis of isolated splenic lymphocytes*—Murine tissue samples were subjected to single-cell purification in complete RPMI 1640 medium (Capricorn Scientific, Ebsdorfergrund, Germany) containing 10% fetal bovine serum (FBS; Sigma-Aldrich, St. Louis, MO, USA) using stepwise filtration through a 100 µm and 30 µm cell strainer (Miltenyi Biotec, Bergisch Gladbach, Germany). An automated Casy TT cell counter (Schaerfe Systems, Reutlingen, Germany) was used to determine final total leukocyte cell counts.

*Proliferative in vitro assays with Dsg3-reactive T cells using [^3^H] thymidine uptake*—A total of 2 × 10^5^ splenocytes were cultured in triplicate in a final volume of 200 µL in RPMI+/+ 10% FCS in 96-well-U-bottom microtiter plates for 3 days at 37 °C in 5% CO_2_. Cultures were stimulated with 10 µg/mL human Dsg3, with a recombinant fragment of type XVII collagen or polyclonally with anti-CD3/CD28 antibodies (0.5 µg/mL Biolegend, San Diego, CA, USA) serving as controls. For the final 18 h, 0.5 µCi [^3^H] thymidine was added to each culture. Cells were harvested, and the samples were measured in a β-scintillation counter, and the data were expressed as [^3^H] thymidine uptake (counts per minute, cpm).

*ELISpot assay*—The enzyme-linked immunospot (ELISpot) assay was performed as previously described, with modifications [29]. Briefly, freshly isolated splenocytes and lymph node cells from Dsg3-immunized HLA-transgenic animals were cultured in 96-well-plates at 1 × 10^6^ cells/mL in RPMI-1640 (Capricorn, Ebsdorfergrund, Germany) supplemented with 10% FCS, 100 U/mL penicillin, 100 μg/mL streptomycin and 2 mmol/L L-glutamine. Cultures were stimulated with 10 µg/mL recombinant Dsg3 or type XVII collagen as a control, while anti-CD3/CD28 stimulation (0.5 µg/mL Biolegend) served as a positive control. On day 3, the cells were transferred to the ELISpot plates, coated overnight at 4 °C with anti-mouse IFN-γ abs (BD Biosciences, San Jose, CA, USA) and incubated for an additional 20 h at 37 °C. Development was performed according to the manufacturer’s protocol (BD Biosciences), and the spots were counted automatically using an ELISpot plate reader (A.EL.VIS, Hannover, Germany).

*Flow cytometric staining and data analysis*—Cells were stained with fluorescently labeled antibodies directly after cell processing. For extracellular staining, samples were incubated for 30 min at room temperature with the following antibodies: aCD3-PerCP-Cy5.5 (17A2), aCD4-FITC (GK1.5), aCD45-PE (30-F11), aIgM-Ep-Cy7 (RMM-1), aCD23-FITC (B3B4) (all BioLegend), aCD19-BV605 (1D3), aCD93-BV421 (AA4.1), aCD21-BV711 (7G6) (all BD) and aCD25-PE (PC61.5; Invitrogen by ThermoFischer Scientific, Waltham, MA, USA). For intracellular staining, cells were fixed and permeabilized using fixation/permeabilization buffer (eBioscience FOXP3/Transcription; Invitrogen) and stained for 20 min at 4 °C using aFoxp3-APC (FJK-16s, Invitrogen by ThermoFischer Scientific, Waltham, MA, USA). Data were acquired on the BD FACS LSR Fortessa equipped with four lasers (BD Biosciences). Standard flow cytometric data analysis was performed using FlowJo version 10.8 (BD Biosciences). Data are displayed as ratios between IVIg/Dsg3 groups and Dsg3-immunized groups (total data per experiment exemplified as displayed in Appendix A).

*Statistical analysis*—Statistical analysis was conducted using Graph Pad Prism version 9. Normality testing was performed using the D’Agostino–Pearson omnibus normality test. For non-normally distributed data and multiple comparisons, the statistical significance was calculated using the unpaired *t* test with Welch’s correction. Significant differences in IVIg-treatment compared to Dsg3-immunized samples were displayed as # (upregulation) or * (down-regulation), respectively.

## 3. Results

### 3.1. Impact of IVIg Treatment on Dsg3-Specific IgG in Immunized HLA-DRB1*04:02-tg Mice

Previous findings showed that humanized HLADRB1*04:02-transgenic mice immunized with human Dsg3 develop a robust T-cell-dependent B-cell response, resulting in the production of Dsg3-specific IgG [25]. We therefore investigated if the application of therapeutic IVIg would alter the T- and B-cell polarization, leading to tolerance induction in a pemphigus mouse model. We established two experimental approaches based on Dsg3 immunization combined with either simultaneous (prevention) (Figure 1A) or subsequent (quasi-therapeutic) (Figure 1B) intraperitoneal application of IVIg. PBS sham-immunized animals served as controls for all experiments (Appendix A). Human serum IgG showed a robust systemic distribution for the duration of the experiments (Appendix A). As expected, serum analysis revealed an increase in Dsg3-specific murine IgG concentration over time (white bars) (Figure 2). Additional IVIg application significantly reduced Dsg3-specific IgG (time points A1 and A2), with the effect lasting up to 47 days after the last treatment (time point A3) (grey bars). While the quasi-therapeutic application also initially revealed a significant reduction in Dsg3-specific IgG (time points B1 and B2), three weeks after the last IVIg treatment, the serum levels of the Dsg3-specific IgG were equal to those of the untreated group (time point B3) (colored bars).

### 3.2. Induction of T Regulatory Cells in Lymphatic Tissue

To analyze the underlying T-cell-mediated tolerance-induced mechanism by IVIg, we next investigated the redistribution of T regulatory cells (Tregs), a subpopulation of T cells essential for immune homeostasis, using flow cytometry in lymphatic cells (Figure 3A). In the spleen, induction of CD4+ CD25+ FoxP3+ Tregs was observed in the prevention setting. In contrast, the quasi-therapeutic approach only showed an initial slight increase (time point B1), while prolonged IVIg treatment did not lead to increased Treg cell numbers (time points B2 and, B3). Only Tregs that were in lymph nodes significantly increased at time point A2. Upon prophylactic IVIg treatment, Tregs were significantly increased at week 5 (B1). At later time points (B2, B3), Treg cell numbers remained unaffected (Figure 3B).

### 3.3. Effects of IVIg Treatment on B-Cell Maturation in Lymphatic Tissue

Since a tolerance-inducing effect of IVIg occurred, as shown in Figure 2, we next aimed to analyze how the maturation of the individual B-cell subpopulations was affected by IVIg. Loss of tolerance by autoreactive B cells is proposed to originate from faulty B-cell development. Therefore, the analysis of CD3− CD45+ CD19+ CD93− B cells was further refined, and they were subsequently identified as CD21+ CD23− marginal-zone B cells and CD21+ CD23+ follicular-zone B cells (Appendix A). Marginal-zone B cells displayed a similar distribution pattern in the spleen and the lymph node (Figure 4). While the ratio of IVIg/Dsg3 to Dsg3 was initially strongly reduced in the IVIg preventive treatment (A1), at later time points (A2, A3), this ratio became inverted. In contrast, the quasi-therapeutic model revealed a strong reduction in marginal-zone B cells with IVIg treatment in both the spleen and the lymph nodes at all time points (B1, B2, B3). Follicular-zone B-cell distribution was less affected by IVIg application. A slight decrease during the course of the treatment was found in the spleen but not in lymph nodes. In the quasi-therapeutic model, both in splenocytes and lymph-node-derived lymphocytes, a similar but weaker trend, however, was observed, as compared to the marginal-zone population.

Transitional-state B-cell subpopulations were defined in the CD3- CD45+ CD19+ CD93+ population using IgM+ CD23− (T1), IgM+ CD23+ (T2) and IgM− CD23+ (T3) (Appendix A). T1 slowly increased over the course of treatment in the prevention groups in both the spleen and lymph node. In contrast, the quasi-therapeutic model displayed a rather stably reduced expression, except at the late time point B3 in the spleen. While T2 in the spleen steadily decreased in the prevention model, the quasi-therapeutic model revealed a consistent up-regulation in both the spleen and lymph nodes. T3 formation also decreased over time in the prevention model, while in the quasi-therapeutic model, an up-regulation was found, especially in the spleen (Figure 5).

### 3.4. Reduced Type I T-Cell Response against Dsg3 upon IVIg Treatment

Since we observed reduced anti-Dsg3 IgG serum levels associated with Treg cell induction in the spleen and lymph nodes, we aimed to further characterize the Dsg3-specific T cells ex vivo. While the proliferation assay based on ^3^H-T incorporation facilitates a more general analysis of T-cell reactivity after Dsg3-specific ex vivo stimulation (Figure 6A), the IFN-ℽ ELISpot enables quantification of the Th1 response pattern (Figure 6B).

As compared to splenocytes derived from the Dsg3-immunized mice (light bars), Dsg3-specific proliferative responses in the IVIg-treated mice displayed a strong decrease upon the preventive treatment (time points A1 and A2) (Figure 6A). No differences were seen at the later time point A3. The quasi-therapeutic protocol only initially displayed a reduced proliferative response upon Dsg3-specific stimulation (B1), while during the course of the experiment (B2 and B3), no difference between the IVIg-treated and untreated mice was evident.

A similar picture was seen after the analysis of IFN-γ-secreting T cells upon in vitro challenge with Dsg3 (Figure 6B). While in the early time points in the prevention protocol (A1, A2), a reduced IFN-γ secretion by the T cells derived from the IVIg-treated mice was observed, further extension of the IVIg treatment to week 10 abolished the Dsg3-specific reactivity (time point A3, Figure 6B). The short-term, quasi-therapeutic protocol revealed, similarly to the cell proliferation, an initial reduction in IFN-γ secretion (B1), whereas the later time points (B2 and B3) only yielded very few or no reactive T cells.

## 4. Discussion

This study provides evidence that protective effects attributed to IVIg during the course of pemphigus treatment can account for both adaptive and innate immunomodulatory mechanisms. We have established an intraperitoneal IVIg treatment regime in a quasi-therapeutic murine PV model displaying an ongoing B-cell-dependent auto-ab formation, similar to IVIg treatment in PV patients. Additionally, we employed a prevention murine PV model which enabled the characterization of IVIg-induced T- and B-cell modulation during the T-cell-dependent onset of Dsg3-specific auto-ab formation. Current concepts strongly suggest that the main mechanisms of IVIg can be divided into antigen-specific and antigen-independent flexible F(ab′)_2_ parts, which mainly covers neutralization of cytokines/cytokine production/anti-idiotypic antibodies. Fc-dependent pathways of IVIg activity include saturation of the IgG-protective neonatal FcR receptor (FcRn), attenuation of complement-mediated tissue damage and effector functional modulation of T and B cells [30]. Here, we were able to show that, in two different settings, a significant Dsg3-specific IgG reduction after IVIg treatment was achieved, accompanied by an induction of Tregs. Furthermore, a distinct reaction profile of T cells via ex vivo stimulation points to reduced Th1 polarization by IVIg. Immature (transitional) and mature (marginal-zone, follicular-zone) B cells displayed a distinct IVIg-induced alteration, pointing to a broad immune regulatory effect by IVIg.

The present study analyzes IVIg-induced tolerizing effects in vivo using transgenic mice that harbor HLA-DRB1*04:02 and HLA-DQB1*03:02 (which is in a linkage disequilibrium with DRB1*04:02) and the human CD4 co-receptor but are devoid of functional murine MHC class II complex (I-Aβ−/−) [25]. As shown by Eming and co-workers, an initial immunization (on day 0) leads to a slow induction of Dsg3-specific IgG, while the second boost on day 14 induces a strong IgG formation that remains stable for well over 10 weeks. Parallel i.p IVIg administration efficiently reduced Dsg3-specific murine IgG, which is also known to occur for other antigens [31]. Interestingly, while human IgG was still detectable after 11 weeks, its tolerizing effect based on the Dsg3-specific IgG levels seemed to weaken 3 weeks after the final IVIg application. This is in line with several human studies that have found only transient improvements in epithelial disorders upon IVIg treatment [32,33].

CD4+ CD25+ FoxP3+ Tregs are integral in balancing immunity and tolerance in various organs, including the skin [34,35]. In both mice and men, crucial IL-2-dependent contribution to tolerance has been well described. Pre-clinical studies have shown that freshly isolated or ex vivo-expanded Tregs can confer immunological tolerance in subjects with autoimmune and inflammatory disorders [36,37,38]. In contrast, removal of Tregs via depletion or loss of function potentially leads to fatal autoimmune diseases, such as type 1 diabetes mellitus [39,40]. Peripheral tolerance depends, among other factors, on the surface factor OX40 on Tregs, which constrains the signaling in autoreactive T cells prior to their disappearance [37] and thereby suggests itself as a therapeutic target for various skin diseases [41]. In the present study, analysis of CD4+ CD25+ FoxP3+ Tregs clearly showed induction in the splenic tissue in the prevention model with IVIg, while this population was nearly unchanged in the quasi-therapeutic model. In the lymph nodes, Tregs were only induced in the short-term model (B1), which reflects a non-lasting, Treg-associated tolerizing effect, displayed in Figure 2.

In addition to the induction of Tregs, which profoundly impacts immune cascades, IVIg has also been shown to inhibit the activation and subsequent production of cytokines by Th1 and Th17 cells in several clinical studies and in vitro experiments [42,43]. ELISpot analysis of splenocyte activity after ex vivo antigen-specific stimulation indeed displayed a reduced Th1 response pattern, quantified as the secretion of IFN-γ by splenocytes derived from IVIg-treated mice in comparison to immunized positive control samples. IVIg can directly shift Th2-type responses [44] by down-regulation of pro-inflammatory cytokines such as IFN-γ. In the current study, later time points, such as A3 (week 10) or B3 (week 11), did not yield any Dsg3-reactive splenocytes with regard to IFN-γ secretion, which underlines the now T-cell-independent auto-ab production by antigen-specific B cells. Our data are in line with studies showing that IVIg stimulates B cells and CD4+ T cells in vivo [45]. Parallel to cytokine production, we investigated the proliferative response of splenocytes after ex vivo stimulation. Using polyclonal aCD3/CD28 stimulation to ensure responsive integrity (Appendix A), similar to the ELISpot analysis, we found that primarily cells from the preventive setup were antigen-responsive, while only the first time point B1 still displayed cellular proliferation. At later time points (B2, B3), T-cell responses became very weak. This investigation demonstrates that primarily during the immunization phase, T-cell activity can be efficiently monitored and influenced with IVIg. Here, rapid ex vivo stimulation assays [46] or the use of novel techniques, such as the application of multimers [47], may provide a detailed characterization of disease-promoting Dsg3-reactive T cells.

The important function of T- and B-cell interaction in PV pathogenesis was supported by a PV mouse model established by Amagai and coworkers [48]. Transfer of splenocytes from Dsg3−/− mice immunized with murine Dsg3 into Dsg3+/+Rag2−/− recipient mice induced a clinical phenotype with mucosal erosions reminiscent of PV. In contrast, depleting CD4+ T cells before transfer into the Dsg3+/+Rag2−/− mice induced neither auto-ab production nor a PV-like phenotype. Similarly, the transfer of B-cell-depleted splenocytes was not sufficient to induce anti-Dsg3 IgG and/or a clinical phenotype. In the present HLA-tg mouse model, abrogation of T- and B-cell interaction with a monoclonal CD40 antibody completely prevented Dsg3-reactive IgG formation. Here, we quantified transitional-state (T1, T2, T3) and marginal- and follicular-zone B cells in lymphatic tissue. Transitional B cells represent a crucial link between immature B cells in the bone marrow and mature peripheral B cells that are potential key players in autoimmune diseases [49,50]. In the preventive approach, we found a sequential rise in T1 cells due to IVIg, at the expense of T2 and T3 cells. While T2 cells can give rise to mature follicular- and marginal-zone B cells, stage T3 B cells represent an anergic population that does not progress into the mature stage but rather is hyporesponsive in the context of autoimmune diseases. It is thus a valid target of immunotherapies based on the interaction with self-antigens [51]. In the quasi-therapeutic approach, however, T1 was constantly reduced, while T2 and T3 were increased in both the spleen and lymph nodes. This points to a different target effect of IVIg on the formation of Dsg3-specific IgG, based on the already T-cell-independent IgG production. Functional ex vivo characterization of the individual transitional B-cell populations might shed more light on the potency of each population with regard to Dsg3-specific IgG secretion after IVIg treatment in vivo.

The surviving immature/transitional B cells subsequently enter or remain in lymphoid tissues and develop further into marginal-zone and follicular B cells. Follicular B cells represent the most common subtype of B cells that are known for their key contribution to adaptive immune responses. They give rise to plasma cells as the terminal differentiation step of mature B lymphocytes and memory B cells. We found a steady decrease in the follicular-zone B cells in the spleen over time in the preventive setup with IVIg application, and, in general, their numbers were lower than those in the Dsg3-immunization-only group. In pemphigus, follicular-zone B cells have been described in the skin as cells that continuously circulate and are rather ineffective in their function as an antigen-presenting cell, yet the disease severity and progression positively correlate with the number of lesional B cells [52]. Marginal-zone B cells were initially strongly reduced with increasing numbers over time in the prevention setup, while in the quasi-therapeutic setup, they were constantly suppressed. These B cells display a characteristic cytokine profile that may enhance autoimmune development [53]. On the other hand, regulatory IL-10, secreted by activated MZ B cells, may contribute to the regulation of systemic autoimmunity in mice. These results suggest the potent effect of IVIg in regulating systemic tolerance. Further studies aiming to dissect the exact effects on the altered functionality of affected B-cell subpopulations are needed.

## Figures and Tables

**Figure 1 cells-12-01340-f001:**
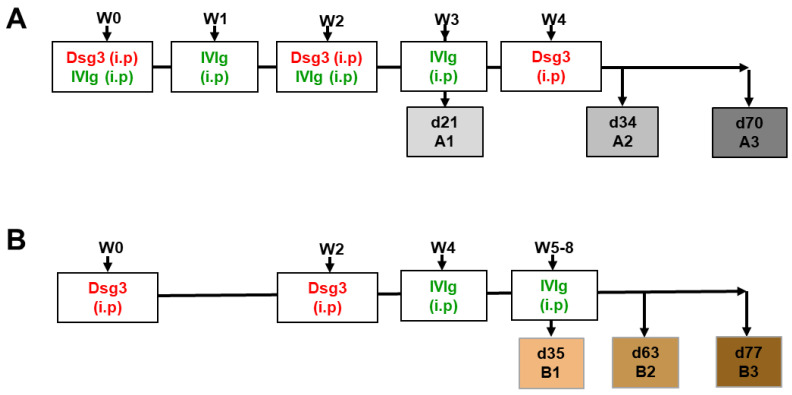
**Schematic display of immunization, treatment and analysis schedule.** (**A**) Prevention model shows IVIg treatment (green) during the onset of antibody formation by Dsg3 immunization (red) at week 0 and week 2 (A1) and week 4 (A2, A3) (grey boxes). (**B**) Quasi-therapeutic model displaying a weekly IVIg application at weeks 4–8, starting at d14 after booster Dsg3 immunization at week 2 (colored boxes). Abbreviations: i.p = intraperitoneal; IVIg = high-dose intravenous immunoglobulin; Dsg3 = desmoglein 3; W = week; d = day.

**Figure 2 cells-12-01340-f002:**
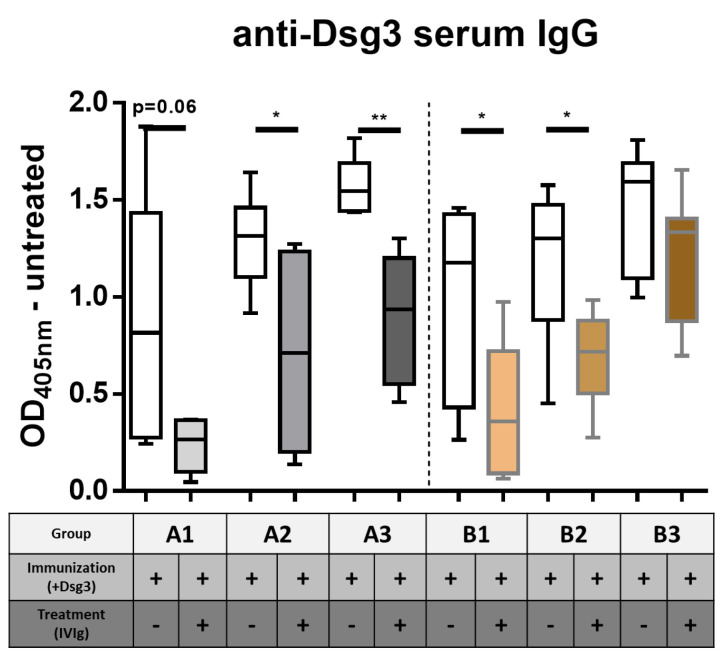
**Serum quantification of Dsg3−specific murine IgG with ELISA.** Statistical significance was calculated using the *t* test with Welch’s correction. *p* ≤ 0.05 *, *p* ≤ 0.01 **. Error bars represent standard deviation, mean ± SD. *n* = 6–8 mice per group, including control PBS groups. A1–3 = prevention protocol, B1–3 = quasi-therapeutic protocol.

**Figure 3 cells-12-01340-f003:**
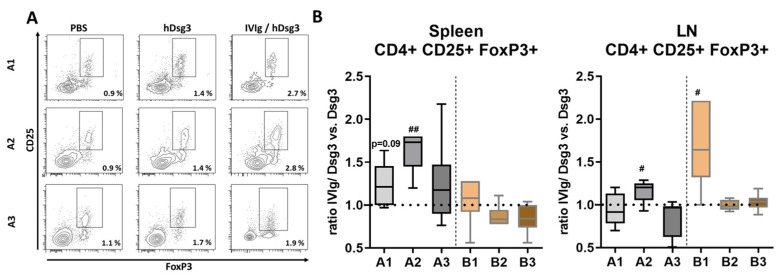
**Flow cytometric analysis and quantification of regulatory T cells in lymphatic tissue.** (**A**) Representative display of flow cytometric gating strategy after preselection on singlets and CD4+ expression. (**B**) Ratio of CD4+ CD25+ FoxP3+ regulatory T cells of IVIg-treated and Dsg3-only immunized mice. Statistical significance was calculated using the unpaired *t* test with Welch’s correction. *p* ≤ 0.05.#, *p* ≤ 0.01 ##. Error bars in graphs represent standard deviation, mean ± SD. *n* = 6–8 mice per group. A1–3 = prevention protocol, B1–3 = quasi-therapeutic protocol. LN = lymph node.

**Figure 4 cells-12-01340-f004:**
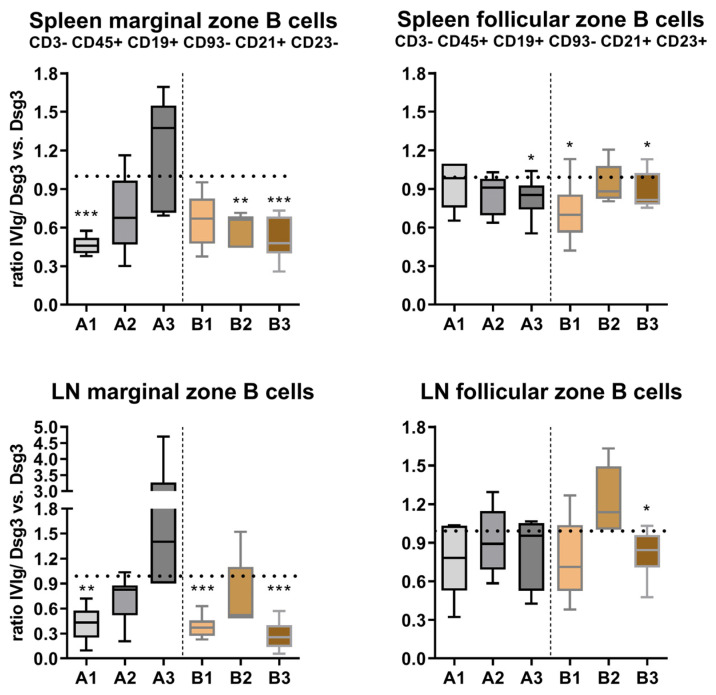
**Analysis of CD3− CD45+ CD19+ CD93− marginal and follicular B-cell subpopulations in lymphatic tissue.** Upper row: analysis of splenic transitional B cells, lower row: lymph node transitional B cells. Statistical significance was calculated using the unpaired *t* test with Welch’s correction. *p* ≤ 0.05 *, *p* ≤ 0.01 **, *p* ≤ 0.001 ***. Error bars in graphs represent standard deviation, mean ± SD. *n* = 6–8 mice per group. A1–3 = prevention protocol, B1–3 = quasi-therapeutic protocol. LN = lymph node.

**Figure 5 cells-12-01340-f005:**
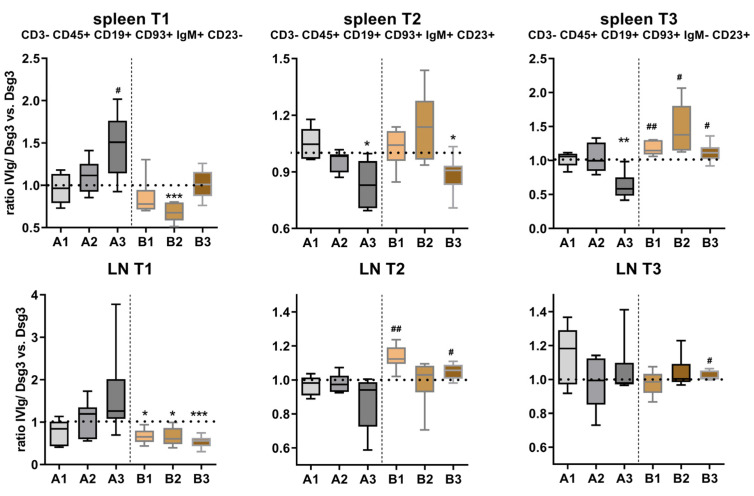
**Analysis of CD3− CD45+ CD19+ CD93+ transitional B-cell subpopulations in lymphatic tissue.** Upper row: analysis of splenic transitional B cells, lower row: lymph node (LN) transitional B cells. Statistical significance was calculated using the unpaired *t* test with Welch’s correction. *p* ≤ 0.05.#, *p* ≤ 0.01 ## (up-regulation or * down-regulation compared to control, respectively). Error bars in graphs represent standard deviation, mean ± SD. *n* = 6–8 mice per group. A1–3 = prevention protocol, B1–3 = quasi-therapeutic protocol. LN = lymph node.

**Figure 6 cells-12-01340-f006:**
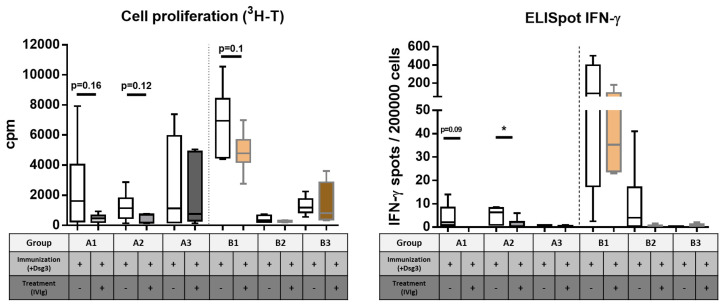
**Impact of IVIg treatment on ex vivo Dsg3-specific proliferation response of splenocytes.** (**A**) ^3^H-T-based cell-proliferation analysis. (**B**) IFN-γ-specific type 1 T-cell response using ELISpot analysis. Statistical significance was performed using the unpaired *t* test with Welch’s correction. *p* ≤ 0.05. (up-regulation or * down-regulation, respectively). Error bars in graphs represent standard deviation, mean ± SD. *n* = 6–8 mice per group. A1–3 = prevention protocol, B1–3 = quasi-therapeutic protocol.

## Data Availability

The raw data supporting the conclusion of this article will be made available by the authors, without undue reservation.

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
