# Peer review of "T Regulatory Cell-Associated Tolerance Induction by High-Dose Immunoglobulins in an HLA-Transgenic Mouse Model of Pemphigus"

_cells, 2023, doi:10.3390/cells12091340_

Round 1

Reviewer 1 Report

Manuscript ID: cells-2359282

Title: T regulatory cell-associated tolerance induction by high dose immunoglobulins in an HLA transgenic mouse model of pemphigus

The manuscript written by Hudemann et al., entitled T Regulatory Cell-Associated Tolerance Induction by High 2 Dose Immunoglobulins in an HLA Transgenic Mouse Model of 3 Pemphigus, relates to  two experimental approaches (preventive and quasi-therapeutic) that examined the potency of IVIg in Pemphigus vulgaris treatment in mice model of the disease.

General:

The sizes of the drawings should be standardized.

Abstract:

Line 29: “Transitional B cells display with regulatory functions, 29 however, their frequency is altered in autoimmune diseases” – sentence is not clear for me.

Line 35-37: “Our data suggest that clinical efficacy 35 of IVIg in the treatment of PV patients is at least modulated by distinct alterations of T- and B cell 36 populations compatible with a tolerance-associated polarization in lymphatic tissue.” The Authors tested their hypothesis on an animal model. Such conclusions are, in my opinion, too far-reaching.

Introduction:

Line 55: Maybe it could be valuable if Authors present more data about PV patients next to the treatment opportunities. How many of them suffer, what are the illness-related symptoms, etc.

Figure 1, line106-111: I do not fully understand the “(A2, A3) (grey boxes)” in this description: “(A) Prevention model shows IVIg treatment (green) during the onset of antibody formation by Dsg3 immunization (red) at week 0 and week 2 (A1) and additionally week 4 (A2, A3) (grey boxes).” Please clarify. The same situation concern the (B) part of the description. The quality of the graphics is low.

Line 113: Human recombinant Dsg3 used for coating ELISA plates was the same that used for mice immunization – extracellular domain of human Dsg3? Please specify.

Line 118: Please specify the conditions: time and temperature of incubations, blocking agent, positive/negative controls, substrate used for ELISA. The description is very brief.

Results:

Figure 2: I would put the description of the x axis, treated the table separately as part B of the drawing. The colures of the bars correlate with the Figure 1 colures concerning sacrifization of an animal? It is unclear regarding the Figure 1 description. Y axis: please reconsider way of writing measured values on the y-axis.

Figure 4, 5: lack of Y axis description in the right-site figures. Maybe addition of A, B, C ets. would be beneficial?

Figure 6: The figure description anticipates A and B part that is not marked. ELISPOT INF-γ part: the Authors used a different notation for the thousandths of a number than in other cases (200.000 vs 12000 in the Cell proliferation part). Tables are the same for both parts as well as for previous Figures. I thing that x-axis should be marked as in Figure 5.

Line 258-259: “Statistical significance was calculated using the unpaired T test with Welch‘s correction. p 258 ≤ 0.05.# (upregulation or * downregulation, respectively).” The notation is unclear to me.

Author Response

Title: T regulatory cell-associated tolerance induction by high dose immunoglobulins in an HLA transgenic mouse model of pemphigus

Ref 1

The manuscript written by Hudemann et al., entitled T Regulatory Cell-Associated Tolerance Induction by High 2 Dose Immunoglobulins in an HLA Transgenic Mouse Model of 3 Pemphigus, relates to two experimental approaches (preventive and quasi-therapeutic) that examined the potency of IVIg in Pemphigus vulgaris treatment in mice model of the disease.

Dear reviewer, thank you for your productive remarks. Please see below our point-by-point review and modified parts in the manuscript cells-2359282R1. 

General:

The sizes of the drawings should be standardized.

Font sizes are now equalized. I will see that in the final pdf from the typesetter the figs. are similar at size (which I cannot influence at initial submission. Fonds are all similar.

Abstract:

Line 29: “Transitional B cells display with regulatory functions, 29 however, their frequency is altered in autoimmune diseases” – sentence is not clear for me.

Sentence changed.

Line 35-37: “Our data suggest that clinical efficacy 35 of IVIg in the treatment of PV patients is at least modulated by distinct alterations of T- and B cell 36 populations compatible with a tolerance-associated polarization in lymphatic tissue.” The Authors tested their hypothesis on an animal model. Such conclusions are, in my opinion, too far-reaching.

I removed “in the treatment of PV patients” to generalize the concept. Since i claim “at least a modulation by IVIg” I think the point of the paper is not too far stretched. Of course, in the end the transition to human cells is desirable, this would however exceed the scope of this manuscript     .

Introduction:

Line 55: Maybe it could be valuable if Authors present more data about PV patients next to the treatment opportunities. How many of them suffer, what are the illness-related symptoms, etc.

Additional information regarding prevalence and clinical symptoms were added to the introduction.

Figure 1, line106-111: I do not fully understand the “(A2, A3) (grey boxes)” in this description: “(A) Prevention model shows IVIg treatment (green) during the onset of antibody formation by Dsg3 immunization (red) at week 0 and week 2 (A1) and additionally week 4 (A2, A3) (grey boxes).” Please clarify. The same situation concern the (B) part of the description. The quality of the graphics is low.

Figure 1 has now been slightly changed so that the analysis time is easier to understand. We have, throughout the manuscript, colour coded the prevention model with grey boxes (A1, A2, A3) and the quasi-therapeutic model with coloured boxes (B1, B2, B3). Hopefully it is now clear. The figures are also uploaded in 600dpi, we believe during the journals conversion process quality is lost. Pls address this issue to the editor if needed.

Line 113: Human recombinant Dsg3 used for coating ELISA plates was the same that used for mice immunization – extracellular domain of human Dsg3? Please specify.

Yes, we work with EC1-5. So agreed, additional information was necessary and added.

Line 118: Please specify the conditions: time and temperature of incubations, blocking agent, positive/negative controls, substrate used for ELISA. The description is very brief.

Additional information was added.

Results:

Figure 2: I would put the description of the x axis, treated the table separately as part B of the drawing. The colures of the bars correlate with the Figure 1 colures concerning sacrifization of an animal? It is unclear regarding the Figure 1 description. Y axis: please reconsider way of writing measured values on the y-axis.

Thanks for that comment. The colours of the groups are equal throughout the manuscript. I was considering colouring the group description in the “table” below, but I believe only having the bars representing respective IVIg treated groups will be sufficient. Since an x-labelling is needed, in my opinion this is the most space saving way to describe each group treatment. (see for example PMID: 29132960).

Figure 4, 5: lack of Y axis description in the right-site figures. Maybe addition of A, B, C ets. would be beneficial?

In order to avoid any further confusion about tissue and b cell subpopulations, we decided to not further break the figures into (A, B etc). Therefore, if having multiple plots besides each other, I believe it is enough to have for example just the y-axis on the left blots since it then addresses all the other blots to the right of it. See for example PMID: 36866120

Figure 6: The figure description anticipates A and B part that is not marked. ELISPOT INF-γ part: the Authors used a different notation for the thousandths of a number than in other cases (200.000 vs 12000 in the Cell proliferation part). Tables are the same for both parts as well as for previous Figures. I thing that x-axis should be marked as in Figure 5.

Thank you for that comment. Changes were made accordingly.

In Fig. 5 the ratio was plotted, therefore only the groups are displayed. In Fig. 6 hDsg3 and hDsg3 / IVIg treated groups are displayed for methodological reasons. Therefore the y-axis in Fig. 6 is now equalized to Fig. 1

Line 258-259: “Statistical significance was calculated using the unpaired T test with Welch‘s correction. p 258 ≤ 0.05.# (upregulation or * downregulation, respectively).” The notation is unclear to me.

The statistics were changed to (hopefully) improve understanding.

Reviewer 2 Report

This study demonstrates that IVIg treatment in a humanized mouse model of pemphigus vulgaris leads to distinct alterations in T and B cell populations, which are compatible with a tolerance-associated polarization in lymphatic tissue.

Minor comments:

Line 45: Please explain PV, because it’s used for the first time (the abstract does not count).

Line 49: Please explain IgG.

Line 96, point 2.2: It is not clearly written, and it does not correlate to fig 1. What is group A and B?

What is the control group?

There is no data about the mice: age, sex.

Fig 1: Please explain “W”. I don’t understand the difference between A2 and A2, the same B1, B2, and B2. Please draw it better.

Line 100: Immunoglobulins  - It's interesting what kind of preparation it is, whether it's human globulins or mice. Please provide a more detailed description. If human, there were no side effects?

Fig 2: The title font is too big. Please adjust.

Author Response

Title: T regulatory cell-associated tolerance induction by high dose immunoglobulins in an HLA transgenic mouse model of pemphigus

Ref 2

This study demonstrates that IVIg treatment in a humanized mouse model of pemphigus vulgaris leads to distinct alterations in T and B cell populations, which are compatible with a tolerance-associated polarization in lymphatic tissue.

Dear reviewer, thank you for your productive remarks. Please see below my point-by-point review and modified parts in the manuscript cells-2359282. 

Minor comments:

Line 45: Please explain PV, because it’s used for the first time (the abstract does not count).

The abbreviation has now been properly introduced in the first paragraph.

Line 49: Please explain IgG.

The abbreviation has now been added in the first paragraph.

Line 96, point 2.2: It is not clearly written, and it does not correlate to fig 1. What is group A and B?

The first part of the results section is slightly rewritten, hopefully it is more understandable now.

What is the control group?

I added an additional sentence in the first part of the results section, so that it is clear that certainly for each experiment a PBS group served as control (see suppl. Fig. 4). Looking at the data (especially Fig. 1) I decided to not depict the pbs group since for anti-Dsg3 there won’t be any signal and the main focus is at the changes (in T and B cells) in IVIg treated Dsg3 immunized mice compared to the “normal diseased” Dsg3 immunized mice.

There is no data about the mice: age, sex.

Apologies, I added a respective part in the M&Ms.

Fig 1: Please explain “W”. I don’t understand the difference between A2 and A2, the same B1, B2, and B2. Please draw it better.

Fig. 1 including legend is now changed for better understanding.

Line 100: Immunoglobulins  - It's interesting what kind of preparation it is, whether it's human globulins or mice. Please provide a more detailed description. If human, there were no side effects?

Intratect is an available human IgG solution used for “Replacement therapy in primary immunodeficiency syndromes (PID) with impaired antibody production and secondary immunodeficiencies (SID) in patients who suffer from severe or recurrent infections, ineffective antimicrobial treatment and either proven specific antibody failure (PSAF) or serum IgG level of < 4 g/l“ (www.biotest.com).

In pemphigus, it works well as a rather cheap substitution therapy (at least temporatily). There are no side effects in mice. Additional information as put in the M&Ms.

Fig 2: The title font is too big. Please adjust.

Figures are now adjusted to each other.

Reviewer 3 Report

In their manuscript, Hudemann and collaborators studied T and B cell polarization by high-dose intravenous immunoglobulins (IVIg) using preventive and quasi-therapeutic models. The authors found that clinical efficacy of IVIgs in the treatment of PV patients is at least modulated by distinct alterations of T- and B cell populations compatible wit a tolerance-associated polarization in lymphatic tissue.

While the manuscript is very well written and the figures and result section are well laid out, the authors should give a bit more context at the beginning of each result section to explain (a sentence or two) why they looked at certain markers or cell types. That would improve the clarity of the manuscript for non-expert readers.

I noticed a couple of mistakes but nothing that makes the text impossible to understand.

Author Response

Title: T regulatory cell-associated tolerance induction by high dose immunoglobulins in an HLA transgenic mouse model of pemphigus

Ref 3

In their manuscript, Hudemann and collaborators studied T and B cell polarization by high-dose intravenous immunoglobulins (IVIg) using preventive and quasi-therapeutic models. The authors found that clinical efficacy of IVIgs in the treatment of PV patients is at least modulated by distinct alterations of T- and B cell populations compatible wit a tolerance-associated polarization in lymphatic tissue.

Dear reviewer, thank you for your productive remarks. Please see below my point-by-point review and modified parts in the manuscript cells-2359282. 

While the manuscript is very well written and the figures and result section are well laid out, the authors should give a bit more context at the beginning of each result section to explain (a sentence or two) why they looked at certain markers or cell types. That would improve the clarity of the manuscript for non-expert readers.

I added more information at the beginning of each section.

I noticed a couple of mistakes but nothing that makes the text impossible to understand.

That is good to hear. Additional proofreading was now performed by external expert.